# Effect of Electric Field on Membrane Fouling and Membrane Performance in Reverse Osmosis Treatment of Brackish Water

**Caixia Fu * , Xuenong Yi and Yuqiong Gao**

School of Environment and Architecture, University of Shanghai for Science and Technology, Shanghai 200093, China; jackyixn@163.com (X.Y.); gaoyq@usst.edu.cn (Y.G.)
* Correspondence: 15921012381@163.com

**Abstract:** One of the most important applied technologies in water treatment is reverse osmosis (RO). However, membrane fouling and flux reduction pose significant challenges. The electric field, as an effective preventive measure, has received limited attention in RO applications. In this study, we added electric fields to finished rolled RO membranes to investigate their effect on membrane fouling and desalination performance. Experimental results indicated that higher electric fields were associated with higher concentrations of treated brine, resulting in a more significant effect. Permeate flux ratios increased with increasing voltage, with peaks of 1.02% (1000 mg/L, 25 V), 1.23% (2000 mg/L, 25 V), and 1.37% (3000 mg/L, 25 V), respectively. Additionally, the maximum reduction in the specific energy consumption (*SEC*) was 31% (2000 mg/L, 25 V) and 59% (3000 mg/L, 25 V), respectively. Notably, electric fields had a retarding effect on $Ca^{2+}$ and humic acid (HA) fouling, with a stronger effect on HA, and higher permeate flux was maintained even after 120 h of operation. While this study visually demonstrates the direct effect of electric fields on RO, further quantification of the economic benefits of this method and a comprehensive understanding of the mechanisms behind how the electric field enhances permeate flux and mitigates membrane fouling are needed.

**Keywords:** reverse osmosis; membrane fouling; desalination performance; permeate flux; electric field

## 1. Introduction

Currently, desalination technology is mainly divided into thermal desalination and membrane desalination. The most widely used thermal treatment technologies are multi-stage flash distillation (MSF) and multi-effect distillation (MED), and the most widely used membrane treatment technology is RO. Compared with MSF and MED technology, RO has obvious advantages, such as the high quality of the produced water, small footprint, low operation cost, and convenient management [1–4]. It is one of the most commonly used methods to solve the shortage of freshwater resources around the world [5,6].

In 2018–2019, countries in the Middle East started to build large desalination projects [7]. As for China, it has been building an average of 10,000 m³/d of large-scale and small/medium projects per year since 2015, with RO dominating projects below 10,000 m³/d [8]. RO technology has accounted for more than half of the global desalination market since 2019 [9].

RO is not only used in large-scale desalination projects but also has many applications in brackish water, sewage, food processing, concentration, and separation [10,11]. Although RO technology has reached a high level of maturity, there are still some challenges to overcome, mainly the problem of membrane pore clogging caused by the high salinity of the feedwater, pollutants, and so on, which in turn lead to a decrease in the osmotic flux of the membrane [1,12,13].

Membrane fouling can be categorized into organic, inorganic, biological, and colloidal fouling, which leads to the formation of salt crystals, cake layers, or biofilms on the membrane surface, resulting in increased transmembrane resistance and reduced permeate

flux [14]. To understand the specific composition of fouling in RO, Li [15] dissected membranes operated during the winter season and found that organic and inorganic fouling exhibited a decreasing trend from the feed end to the permeate end; organic fouling was dominated by polysaccharides, proteins, and biological contaminants; and inorganic fouling was primarily composed of Al, Ca, Mg, Na, and K. It was also observed that organic fouling was more susceptible to microbial fouling. Yokoyama [16] proposed a scaling-based flux model to explain the scaling mechanisms in RO membranes: an increase in solute flux leads to supersaturation and scaling deposition; as the scaling layer thickens, the hydraulic resistance increases, reducing solute transport and decreasing supersaturation; when the scaling layer reaches a steady state, the hydraulic resistance stabilizes and the permeate flux through the membrane no longer decreases. Tong [17] pointed out that mineral scaling is affected by the characteristics of the feedwater: different chemical properties in the feedwater lead to the formation of different types of mineral scaling, and the concentration polarization and concentration difference within the feedwater can both enhance the likelihood of scaling formation. Additionally, Chiao [18] discovered that ions can interact with each other, such as the interaction between calcium ions and alginate, forming a gel network that tends to aggregate near the membrane surface, creating a relatively dense fouling layer.

Ruiz-García's [19] simulation study of the RO treatment of groundwater with minerals such as calcium and silica as the main constituents showed that relatively high flow recovery (R) could only be achieved with the addition of a scale antiscalant at the desired energy consumption, but higher R values (72%) required the addition of a special silica antiscalant, which was approximately twice as expensive as the normal scale inhibitor. Pearson [20] investigated the economics and energy consumption of RO desalination technology for brackish water and found that controlling the concentration of calcium, iron, and silicon in the feedwater is crucial to prevent scaling on the membrane surface, with calcium carbonate and iron scaling mitigated by adjusting the pH. However, the cost of pretreatment to remove calcium sulfate and dissolved iron is relatively high, and silicon scaling proves more difficult, so prevention becomes the primary focus.

In summary, membrane fouling can have negative effects on the treatment and operational costs of RO systems, leading to a decrease in water quality and a shorter lifespan for the membranes. Additionally, it increases the maintenance and replacement costs of the membranes and raises the risk of biofouling. Therefore, the development of effective strategies to mitigate membrane fouling and enhance permeate flux continues to be the primary objective and challenge in current research.

To enhance the efficiency of membrane desalination, researchers often explore various factors, including membrane materials, operating conditions, and influent conditions [21]. Pretreatment, backwashing, membrane surface modification, and chemical treatment are commonly studied approaches in this regard [22–24]. However, such methods may bring additional costs, secondary pollution, and potential physical damage to the membrane structure. Moreover, membrane modification and chemical treatment often involve lengthy research and development cycles. As an alternative, the integration of electric or magnetic fields with RO technology has emerged as a promising avenue to improve membrane desalination performance while mitigating some of these concerns. This innovative approach offers potential benefits in terms of enhanced efficiency and reduced operational and environmental impacts, making it a compelling area of investigation for researchers in the field.

Research on electric fields to reduce membrane fouling and improve permeate flux is not new; there have been many studies on electric field application in microfiltration, ultrafiltration, nanofiltration, and forward osmosis, and some success has been achieved [25]. Du [26] studied the impact of $Ca^{2+}$ on the fouling of the membrane in electric field-assisted microfiltration filtration and found that the charge shielding effect of $Ca^{2+}$ was obvious at 1.5 V, which meant that the slowing down of membrane fouling by the electric field was not obvious. When it was increased to 3 V, the assisting influence of the electric field

became apparent. Yin [27] introduced a self-generated electric field into the MBR system, operating at a maximum electric field intensity of 11.83 mV/cm; because of the action of the electric field-generated $H_2O_2$ and $\cdot OH$, the surface pollutant-specific area on the membrane was reduced by 68.2% compared with the control group. Hu [28] investigated the effect of electro-ultrafiltration membranes on the antiscaling of natural organic matter (NOM) and also concluded that the electric field caused the ultrafiltration to increase the flux of hydrophilic substances in NOM by 20%, and the improvement in the solution with the coexistence of $Ca^{2+}$ and HA was more obvious. Xu [29] showed a conductive thin-film composite forward osmosis membrane, which, at a voltage of 2 V, was highly effective in preventing organic substances from adhering to the surface of the membrane. In addition, some have used the direct combination of an electric field and forward osmosis technology to alleviate the fouling problem during algae harvesting, and it was found that the electric field was effective in reducing the formation of algal fouling on the membrane surface. When a higher electric field was added, the permeate flux increased by 20–40% and the recovery increased by 10–20% [30].

Cao [31] used a variable-frequency electric field as a pretreatment process for seawater treatment by RO, and the solute rejection and permeate flux were increased by 15–35%, respectively, and the pollutants adhering to the membrane surface were also relatively reduced. Similarly, Penteado de Almeida [32] applied an AC-induced electromagnetic field (EMF) to RO wastewater treatment. The system achieved a 13% increase in the recovery rate, along with a 2–8 times reduction in the scaling rate. Moreover, the effect of EMF was stronger when it was used simultaneously with the antiscaling agent, and the recovery rate of the system reached 89.3%. There are also a few researchers who have used the electric field directly on the RO, using the electric potential as a tool to prevent biofouling; when biofouling was present, it was shown that the RO was subjected to potentiodynamic polarization for 30 min and was able to recover 33–44% of the osmotic flux [33].

In summary, the application of electric fields is highly beneficial in preventing the fouling of filtration membranes with pollutants and increasing the permeate flux through the membranes. However, there remains a paucity of research on the direct impact of electric fields on RO desalination, particularly on the direct incorporation of electric fields into completed RO membranes. Consequently, this study aims to directly introduce electric fields to finished roll membranes to examine their influence on RO desalination performance, as well as their effect on both inorganic and organic fouling. The findings from this research will contribute to a deeper comprehension of the mechanisms by which electric fields affect the performance of RO systems, offering valuable insights into the implementation of electric fields in completed RO operations.

## 2. Materials and Methods

### 2.1. Materials

A ULP3012-400 RO membrane with a solute rejection level of 95% was used, which was purchased from Jinan Dekun Water Treatment Equipment Co., Ltd., Jinan, China. The titanium-coated ruthenium anode plate and stainless steel cathode plate were purchased from a local supplier. The DC power supply used, purchased from Jiangsu Ritai Environmental Protection Engineering Co., Ltd., Yancheng, China, varied from 0 V to 50 V.

The chemical reagents used in the experiments, including NaCl, $CaCl_2$, HA, HCl, and NaOH, were analytical-grade reagents purchased from Sinopharm Chemical Reagent Co., Ltd., Shanghai, China. The organic pollutant used was HA, which was purchased from Tianjin Balance Biotechnology Co., Ltd., Tianjin, China.

### 2.2. Test Equipment

The equipment utilized for testing in this study was an RO membrane that was modified with the addition of an electric field, resulting in what is known as Multi-field reverse osmosis (MFRO for short). The MFRO system comprised several features, including an RO membrane, a PP membrane case, a titanium-coated ruthenium anode plate, a

stainless steel cathode plate, and terminals. The cathode plate was secured firmly inside the membrane case near the water outlet using stainless steel screws, while the anode plate was fixed inside the membrane cover near the water inlet. The electric field generated by the anode and cathode plates was aligned in the same direction as the water flowed through the membrane. These plates were square-shaped and measured $50 \times 50 \times 3$ mm, and the outer shell of the MFRO unit was 90 mm in diameter and it had a height of 358 mm. Details on the MFRO unit and the testing equipment are depicted in Figure 1.

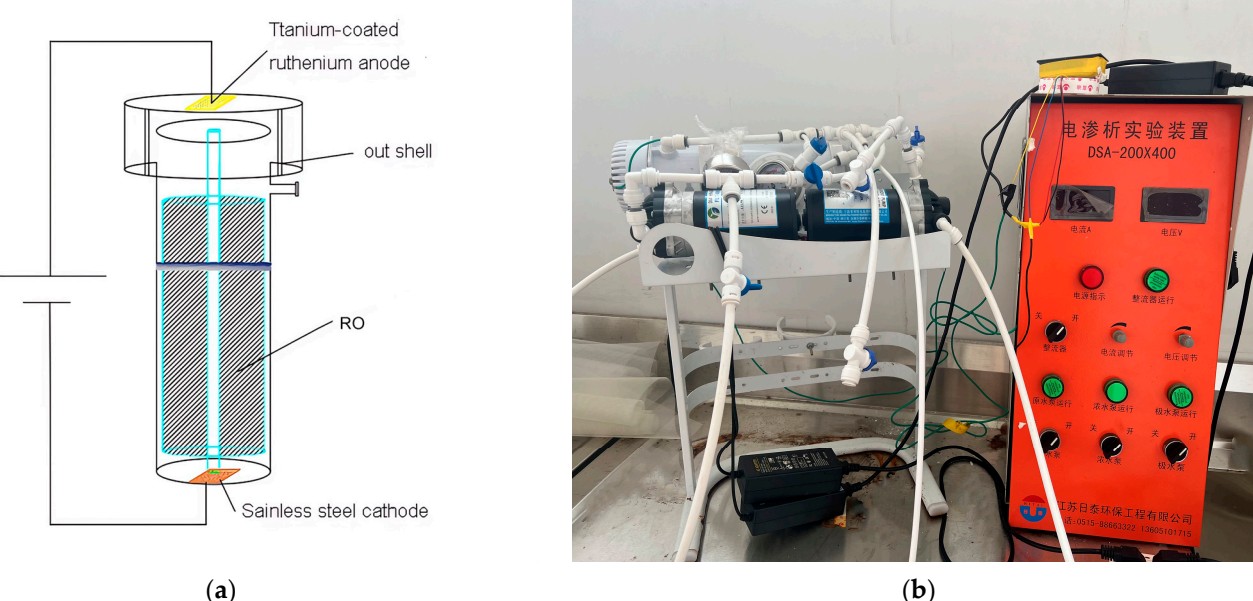

(**a**) (**b**)

**Figure 1.** Structure of testunit (**a**) and physical diagram of test equipment (**b**).

The experimental setup for this study comprised several components, including a raw water tank, booster pump, MFRO unit, pressure gauge, DC power supply, inlet, and outlet pipes. The test water was pumped into the MFRO unit using a booster pump, with the concentrated water and desalinated water flowing out of the concentrated water and fresh water pipes, respectively. The inlet flow rate was regulated by the flow meter and inlet valve, while the pressure was carefully controlled through the back pressure system. Figure 2 displays the schematic diagram of the process flow employed in this study.

### 2.3. The Impact Mechanism of Electric Field on RO

Electric fields' impacts on membrane filtration systems include electrodynamics and electrochemistry, the most important of which are electrophoresis, electrical permeability, and electrostatic repulsion effects [34,35]. During operation, a boundary layer is generated near the membrane, and the concentration of ions, molecules, or particles near the boundary layer will be relatively high. They can be easily deposited on the surface of the membrane when forming a filter cake layer, affecting the desalination performance of the membrane. Since particles such as colloids and ions are generally negatively charged, the addition of an electric field creates a force opposite to that of the membrane surface, which keeps the material away from the membrane surface, and it is carried away by the mainstream liquid, thereby reducing membrane fouling [35]. The equation that defines the thickness of the boundary layer after applying an increased electric field can be derived by combining the N-S equation with Prandtl's boundary layer theory, as shown in [36]:

$$\delta = \frac{4.0652\mu}{v + v_1} \tag{1}$$

where $\mu$ is the kinematic viscosity of the liquid, $v$ is the velocity generated by the operating pressure, and $v_1$ is the velocity generated by the action of the electric field.

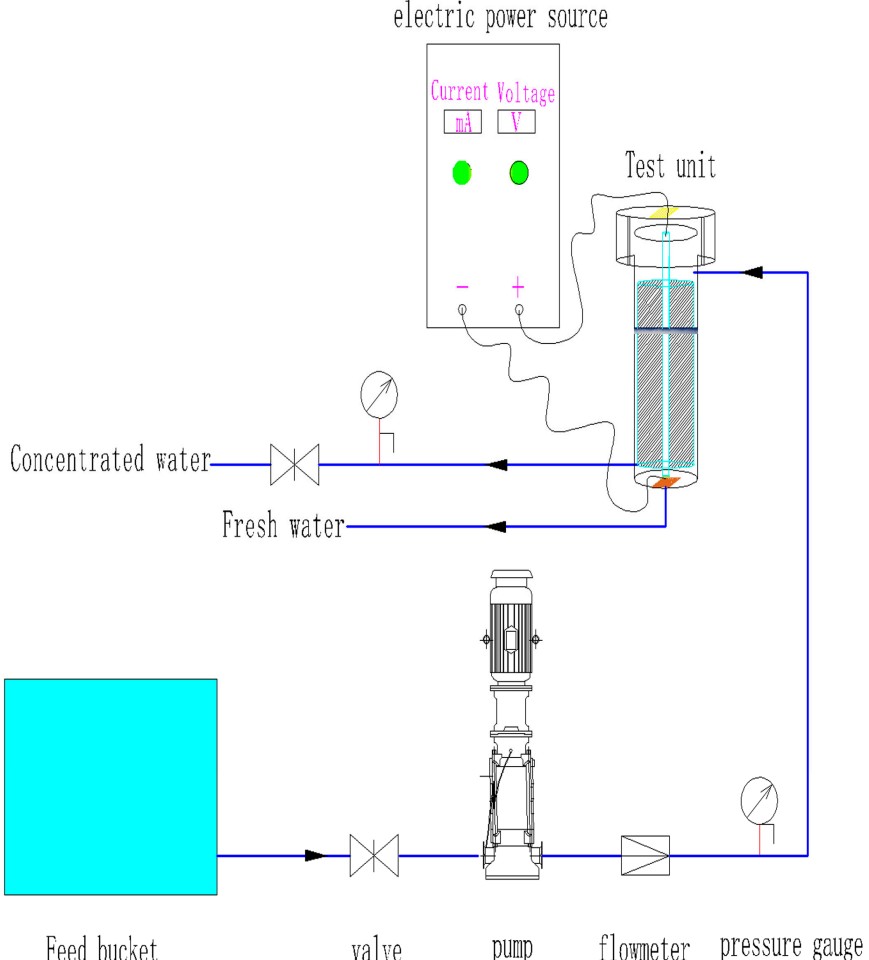

**Figure 2.** Flowchart of the brackish water experiment.

Reducing the thickness of the interfacial layer can alleviate the concentration polarization phenomenon and also reduce the membrane pore blockage caused by the attachment of ions to the membrane surface, which in turn can prolong the membrane lifetime and improve the permeate flux. The mass transfer process of boundary layer particles in the presence of an electric field is shown in Figure 3 [37].

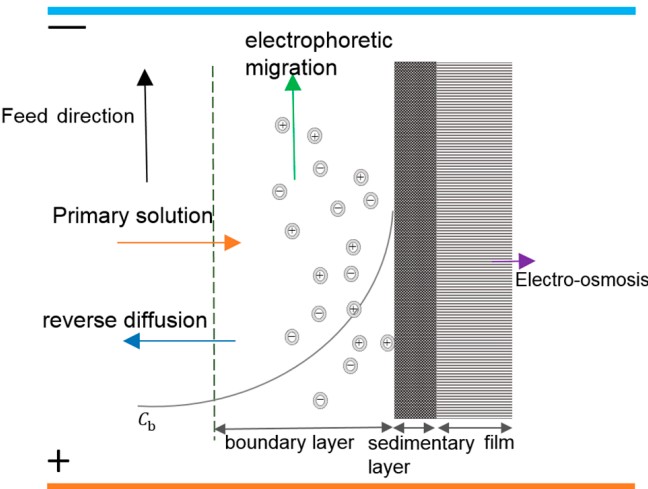

**Figure 3.** Mass transfer process of boundary layer particles in the presence of an electric field.

### 2.4. Steps in the Implementation of the Experiment

This study was divided into three independent experiments. To ensure the reliability of the three independent experiments at the time of testing, each test used a new membrane. The water samples were manually prepared, using HCl and NaOH to control the pH of the sample in the range of $7.0 \pm 0.3$, and the temperature was controlled in the range of $25 \pm 2\,^{\circ}$C. Each test was designed to flow at 1.5 L/min. The system was allowed to stabilize for 30 min before starting each set of tests.

Table 1 shows the MFRO desalination performance test, Table 2 shows the test of the electric field's effect on the RO permeate flux and solute rejection, and Table 3 shows the membrane fouling test.

**Table 1.** MFRO desalination performance test.

| No. | Feedwater Total Dissolved Solids (TDS) (mg/L) | Operating Pressure (MPa) | Sampling Methods |
|---|---|---|---|
| | NaCl | | |
| 1 | 1000 | | Samples were taken at 15-min intervals, with each sample taken three times in parallel. |
| 2 | 2000 | 0.36, 0.45, 0.5, 0.6 | |
| 3 | 3000 | | |

**Table 2.** Test of electric field's effect on the RO permeate flux and solute rejection.

| No. | Feedwater TDS (mg/L) | Operating Voltage (V) | Operating Pressure (MPa) | Sampling Methods |
|---|---|---|---|---|
| | NaCl | | | |
| 1 | 1000 | | | Samples were taken at 15-min intervals, with each sample taken three times in parallel. |
| 2 | 2000 | 0, 5, 10,15, 20, 25 | 0.5 | |
| 3 | 3000 | | | |

**Table 3.** Membrane fouling test.

| No. | Feedwater TDS (mg/L) | | | Operating Voltage (V) | Operating Pressure (MPa) | Sampling Methods |
|---|---|---|---|---|---|---|
| | NaCl | CaCl$_2$ | HA | | | |
| 1 | 3000 | 200 | 0 | | | Samples were taken at 8-h intervals and each sample was taken three times in parallel for a total working time of 120 h. |
| 2 | 3000 | 0 | 20 | 10 | 0.5 | |
| 3 | 3000 | 0 | 0 | | | |

### 2.5. Analytical Methods

Throughout the test, it was imperative to monitor the feed, concentrate, and purified water flow rate and quality, as well as the pressure before and after the RO membranes. The flow was monitored using an on-line flow meter and the pressure was monitored using an on-line pressure gauge. A portable conductivity meter (model DDBJ-350, Yantai Stark Instrument Co., Ltd., Yantai, China) and a pH meter (model pH-009(I) A) were utilized to measure the TDS, pH, and temperature. The tests were carried out using configured water samples, keeping the pH and temperature constant.

After each condition of the membrane fouling experiments, the RO membranes were disassembled. Subsequently, 1 cm $\times$ 1 cm samples were temporarily stored in a refrigerator for later analysis. At the end of the entire membrane fouling experiment, these samples were scanned using a scanning electron microscope (Hitachi SU810, Tokyo, Japan) at a maximum magnification of 500,000$\times$.

*2.6. Calculations*

To assess the desalination performance of the retrofitted MFRO system, commonly utilized metrics such as the permeate flux *J*, solute rejection Ø, permeate flux ratio *J*%, and recovery rate *R*, along with the specific energy consumption (*SEC*), were employed. The calculation of these indices, which related to the system as a whole, was based on data derived from parameters such as the inlet water salinity, produced water salinity, inlet pressure, outlet pressure, and current and voltage measurements. These data provide the basis for accurate evaluation and analysis.

The permeate flux (*J*, L/(m²·h)) is defined as the volume of water that passes through one square meter of the reverse osmosis membrane per hour.

$$J = \frac{V}{S \times t} \tag{2}$$

where *V* is the water permeation volume at time *t* (L), *S* is the effective membrane area of 1.8 m², and *t* is the sample collection time (h).

Solute rejection refers to the extent of the difference in the solute concentration between the permeate and the feedwater when subjected to the RO membrane. It serves as a measure of the membrane's effectiveness in removing solutes.

$$= \frac{C_0 - C_T}{C_0} \times 100\% \tag{3}$$

where $C_0$ refers to the solute concentration (mg/L) in the permeate, while $C_T$ represents the solute concentration (mg/L) in the feedwater.

The permeate flux ratio (*J*%) refers to the ratio of the permeate flux at time *t* to the initial permeate flux:

$$J_\% = \frac{J_{vt}}{J_{vt0}} \tag{4}$$

where $J_{vt0}$ refers to the initial moment of water flux (L/m²·h, liters per square meter per hour); and $J_{vt}$ refers to the water flux at time *t* (L/m²·h).

The recovery rate (*R*) is defined as the percentage of water infiltrated relative to the total amount of water treated by the system after time *t*.

$$R = \frac{V}{V_0} \times 100\% \tag{5}$$

where $V_0$ is the total volume (L) of feedwater that has been treated during *t* hours of operation.

*SEC* [38,39] represents the amount of energy required to produce one liter of water, expressed in kW·h/L.

$$SEC = \frac{W}{V/t} \tag{6}$$

$$W = \frac{P_{wp}}{R} + W_E \tag{7}$$

$$W_E = \frac{U \times I \times T_1 \times \eta}{1000} \tag{8}$$

where *W* represents the overall energy consumption of the treatment system per hour (kW·h), $P_{wp}$ is the operating pressure provided by the booster pumps (MPa), $W_E$ is the energy consumption to produce one ton of water for the electric field (kW·h), *U* refers to the operating voltage of the system (V), *I* refers to the operating current of the system (A), $T_1$ refers to the treatment time (1 h), and $\eta$ refers to electrical energy conversion efficiency, taken as 0.8.

## 3. Results

### 3.1. Influence of the MFRO on the Desalination Efficiency in Purified Salt Water

The desalination performance of the MFRO equipment under various salt concentrations is illustrated in Figure 4. Figure 4a shows the solute rejection, while Figure 4b represents the recovery rate. The highest solute rejection achieved for salt concentrations of 1000 mg/L, 2000 mg/L, and 3000 mg/L was 97.37%, 95.08%, and 93.46%, respectively. Meanwhile, the maximum recovery rates were 32.55%, 20.82%, and 16.34%, respectively. In general, lower salt concentrations corresponded to higher desalination and recovery rates. This can be attributed to the lower osmotic pressure and weaker ion interaction in low-salinity solutions, resulting in less concentration polarization. With an increasing ion concentration in the feedwater, the membrane pressure and surface potential changed, leading to increased concentration polarization and potential ion precipitation, which served to obstruct the membrane pores and subsequently reduce the permeate flux and solute rejection. However, beyond a certain range of salinity, where the ion concentration surpasses the membrane limit, the effect on membrane filtration selectivity becomes negligible. In such cases, the solute rejection stabilizes, and no significant changes occur [40,41].

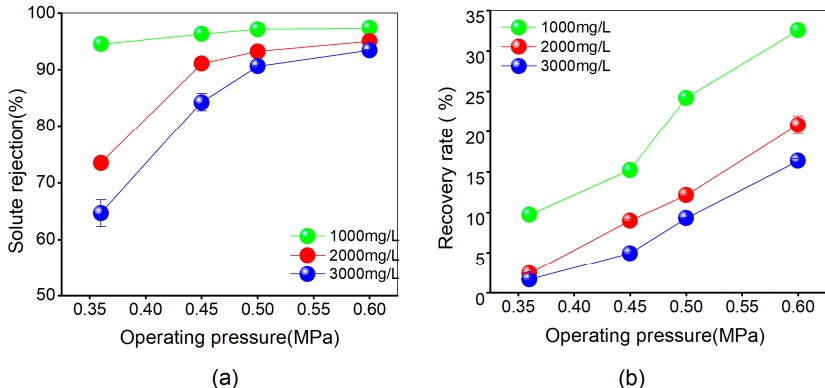

(a)                  (b)

**Figure 4.** Comparing membrane performance of the Multi-field reverse osmosis at different pressure fields under pure brine conditions: (**a**) changing the solute rejection and (**b**) changing the recovery rate.

The research results indicate that the solute rejection and recovery trends of the MFRO equipment under different salt concentrations and operating pressures are similar to those in other scholars' studies, conforming to the principles of traditional RO systems. These results confirm the applicability of MFRO equipment for future experimental investigations.

### 3.2. Influence of Electrical Field on the Performance of MFRO Desalination

To study the effect of an electric field on the rate of desalination, the flux through the membrane, and the *SEC* in an RO system, the operation pressure was maintained at a fixed value of 0.5 MPa, while the voltage magnitude was systematically varied. The permeate flux impact was analyzed by calculating the permeate flux ratio using Equation (4).

Figure 5a illustrates that irrespective of the specific concentration, there was no discernible change in solute rejection following the application of the elevated electric field over the case with no electric field. This result shows that at salt concentrations below 3000 mg/L, the electric field has no significant effect on the desalination of the membrane.

Figure 5b shows that the permeate flux ratio increased slightly when an electric field was applied. The values recorded were 1.02%, 1.23%, and 1.37% for salt ion concentrations of 1000, 2000, and 3000 mg/L, respectively. It should be noted that the effect of the electric field on the permeate flux was more pronounced at higher salt ion concentrations and increased voltages, eventually reaching a plateau. According to Equation (2), the amount of produced water is directly influenced by the permeate flux, indicating that water production increases with higher voltages. While water production is typically affected by the operating pressure within the normal range, our study maintained a constant pressure and only varied the electric field. These findings suggest that the electric field can enhance

ion migration, reduce ion accumulation on the membrane surface, promote water molecule movement, and thus increase the permeate flux.

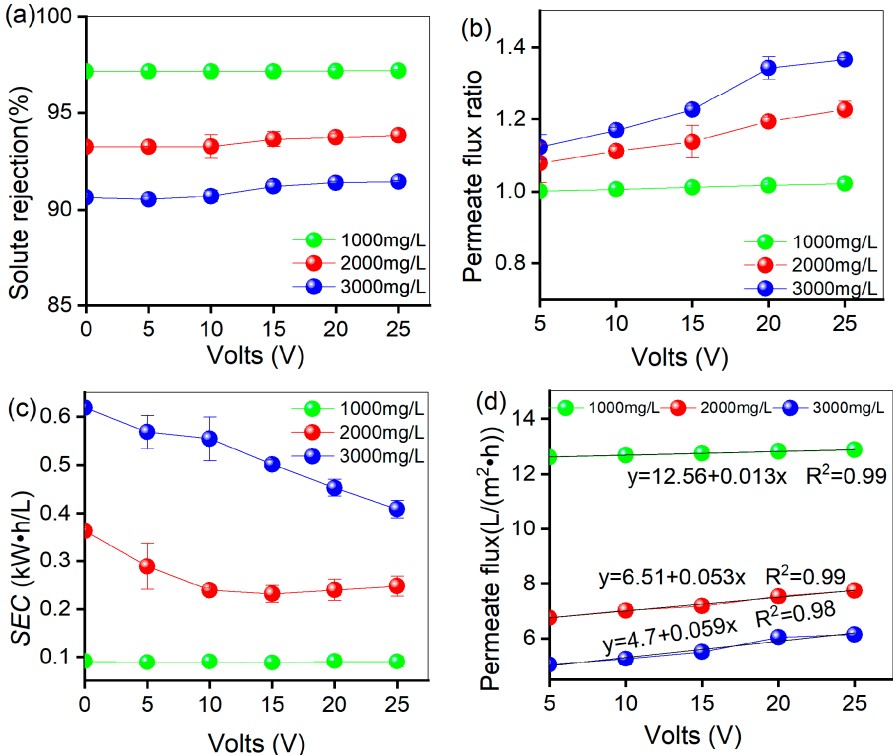

**Figure 5.** Under pure brine conditions, an electric field was added to compare the membrane performance of the Multi-field reverse osmosis under different voltage conditions: (**a**) changing the solute rejection, (**b**) changing the permeate flux ratio, (**c**) changing the *SEC*, (**d**) linear fit of permeate flux to voltage.

Figure 5c shows that the *SEC* in the presence of an electric field is lower at different concentrations than in the absence of an electric field. Specifically, when the salt concentrations are 2000 and 3000 mg/L, the *SEC* decreases from 0.36 and 0.62 kW·h/L (0 V) to 0.25 and 0.39 kW·h/L (25 V), respectively. These reductions correspond to a decrease of 31% and 59%, respectively. However, at a salt concentration of 1000 mg/L, the electrical field has no significant effect on energy consumption. Equations (6)–(8) show that in low-salinity solutions, the current is small, so the electric field's effect on the permeate flux is small. As a result, the rates of power consumption and permeate flux offset each other, leading to a minor change in the unit energy consumption of water production. As the salt concentration increases, the electric field force strengthens, thereby enhancing the effect of the electric field on permeate flux. The growth rate of the permeate flux becomes dominant, leading to a downward trend in energy consumption for water production. However, due to the progressive increase in voltage, the growth rate of power consumption becomes dominant, thereby slowing down the rate of decline in unit energy consumption. Overall, these results suggest that the use of an electric field can greatly decrease the *SEC*, especially at higher salt concentrations.

To investigate the correlation between the permeate flux and operating voltage, a linear regression analysis was conducted. The results, presented in Figure 5d, indicate a strong linear correlation between permeate flux and the applied voltage, with $R^2$ values above 0.9 for all three concentrations tested. The data also reveal that the influence of the voltage on permeate flux varies with the salt concentration. At 3000 mg/L, the permeate flux is most influenced by the applied voltage, followed by 2000 mg/L and then 1000 mg/L. The increase in permeate flux with the voltage was relatively small at a salt concentration of 1000 mg/L, but the increase in permeate flux gradually increased with increasing salt

concentrations. This suggests that the electric field force has a more significant impact at higher salt concentrations. One possible reason for this observation is that the electric field force can affect the arrangement and orientation of water molecules. This leads to the creation of a driving force that accelerates the movement and diffusion of water molecules, facilitating their passage through the semipermeable membrane. Additionally, concentration polarization at the membrane surface is reduced by ionic electromigration induced by the electric field. Particularly in high-salinity water, the salt ions are highly concentrated and prone to polarization, which inhibits water permeation. The electric field effectively overcomes these effects, thereby increasing the permeate flux of the RO membrane. This effect becomes more pronounced with higher salt concentrations.

### 3.3. Effect of $Ca^{2+}$ or HA on MFRO Permeate Flux under Electric Field Conditions

In this study, we investigated the fouling of the RO membrane by using common contaminants found in water treatment applications, including $Ca^{2+}$ and HA.

Figure 6a demonstrates that after 88 h of continuous operation, the permeate flux dropped from 5.49 L/(m²·h) to 4.51 L/(m²·h) and then reached a steady state. However, in the presence of an electric field, the permeate flux was still on a downward trend, the rate of decline decreased, and, after 120 h of operation, it approached the level observed without the electric field. The reduction in permeate flux can be attributed to the presence of calcium chloride ($CaCl_2$) in the water, which dissociates into $Ca^{2+}$ and $Cl^-$ ions. These ions influence the effective size and charge density of other ions in the solution, leading to their adhesion to the membrane surface. Consequently, fouling occurs, resulting in a decrease in the permeate flux. However, when the electric field intensity is increased, it induces changes in the velocity and trajectory of the $Ca^{2+}$ ions, thus reducing their residence time on the surface of the membrane. Additionally, applying the electric field creates a fluid mixing effect that helps to reduce the fouling of the RO membrane.

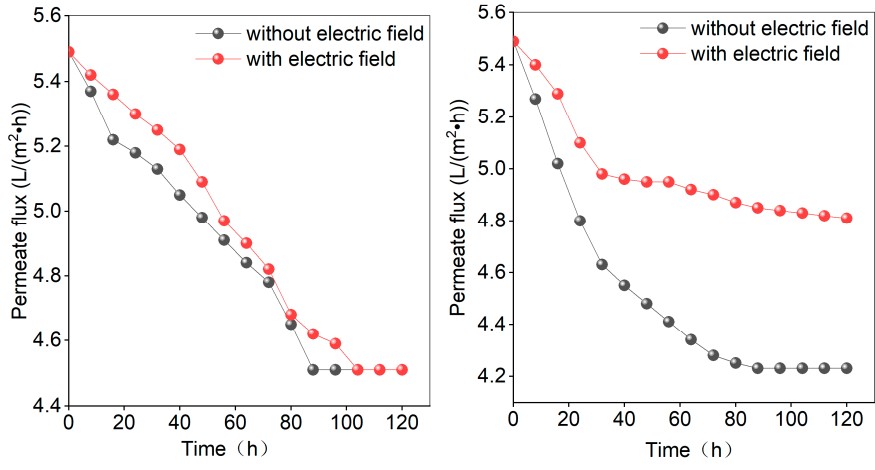

(a) Effect of $Ca^{2+}$ pollution on permeate fluxes    (b) Effect of HA pollution on permeate fluxes

**Figure 6.** Effect of $Ca^{2+}$ or HA on the Multi-field reverse osmosis permeate flux in the absence of an electric field.

Figure 6b illustrates the effect of HA on permeate flux with and without an applied electric field. In the absence of an applied electric field, the permeate flux shows an initial rapid decrease, followed by a slower decrease until it finally stabilizes. From the beginning, the permeate flux decreases from 5.49 L/(m²·h) to 4.23 L/(m²·h). The decrease can be attributed to the dissolution of the HA in the water, as it begins to attach and adsorb to the surface of the membrane, leading to the fouling and clogging of the membrane. As time progresses, the degree of membrane fouling gradually increases, but the permeate flux is ultimately influenced by the RO membrane's stability and approaches a balanced state. When an electric field exists, the rate of permeate flux reduction slows down, and, after 24 h of operation, it reaches a plateau. Eventually, the permeate flux stabilizes at 4.81 L/(m²·h),

indicating that the electric field reduces membrane fouling. This can be attributed to the attraction between HA and the anisotropic charge in the solution when acted upon by the electric field. Simultaneously, it becomes more difficult for pollutants to adhere to the membrane surface, reducing fouling, which is also related to electrophoretic migration and electrostatic repulsion [28,42]. In the absence of other pollutants, the assisting effect of the electric field gradually stabilizes over time.

### 3.4. Effect of $Ca^{2+}$ or HA on MFRO Solute Rejection under Electric Field Conditions

According to the data presented in Figure 7a, the solute rejection was reduced in the presence of $CaCl_2$, dropping from an initial 90.63% to 87.69% without the electric field and to 88.25% with it. This can be attributed to the different ions that accumulate on the surface of the membrane, causing an increase in the concentration polarization phenomenon and a subsequent decrease in the salt permeability rate, ultimately reducing solute rejection. However, when the electric field is applied, $Ca^{2+}$ migrates directionally, resulting in a relative decrease in the concentration of ions remaining near the membrane and the weakening of the concentration polarization. Consequently, the rejection of solutes is reduced.

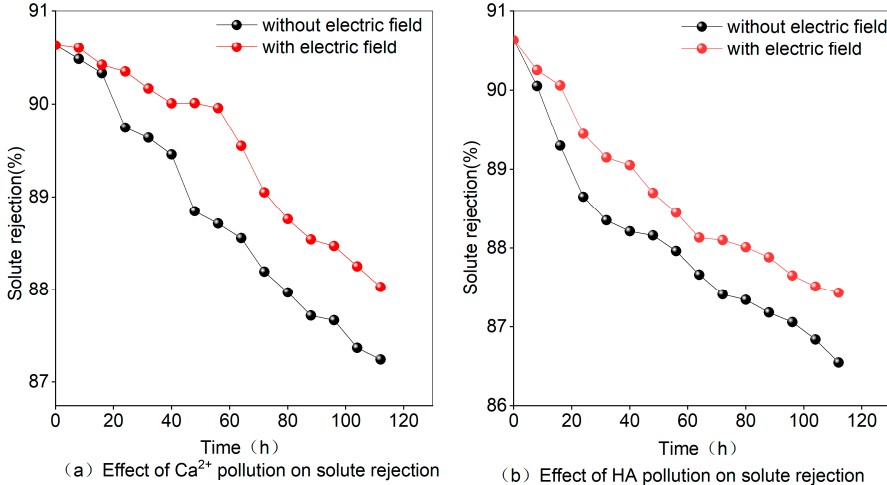

**Figure 7.** Effect of $Ca^{2+}$ or HA on the Multi-field reverse osmosis solute rejection in the absence of an electric field.

Figure 7b shows that the presence of HA also reduces solute rejection, from an initial value of 90.63% to 86.55% without an electric field and to 87.43% with it. The rate of decline is faster in the initial stages and tends to level off in the later stages. This can be explained by the adsorption effect of HA, which initially leads to the direct plugging of membrane pores and a reduction in permeate flux, ultimately affecting solute rejection. As the HA deposits become thicker and reach the membrane limit, the pollution tends to stabilize. However, the addition of an electric field leads to electrophoretic movement, causing HA to combine with ions in the water and leading to a slower rate of decline compared to the case without an electric field.

Figure 7b illustrates that the presence of HA reduces the efficiency of solute rejection in RO membranes. Under normal conditions without an electric field, the solute rejection rate decreases from an initial value of 90.63% to 86.55%. When an electric field is applied, the solute rejection rate decreases to 87.43%. HA directly blocks the membrane pores, leading to a decrease in permeate flow and a subsequent effect on solute rejection, especially in the initial stages. As HA deposits accumulate and reach the membrane limit, the fouling stabilizes. However, the application of an electric field induces electrophoretic movement, causing HA to combine with ions in the water. This slows the rate at which solute rejection decreases compared to the non-electric field case, alleviating the contaminating effect of HA on the RO membrane.

### 3.5. Membrane Fouling Characterization

To more accurately characterize membrane fouling after the addition of an electric field to RO, SEM was employed for membrane fouling characterization.

Figure 8a presents the condition of a new membrane, showing a smooth surface free from pollutants. Figure 8b–e depict the membrane surface after the addition of pollutants, indicating the presence of pollutants.

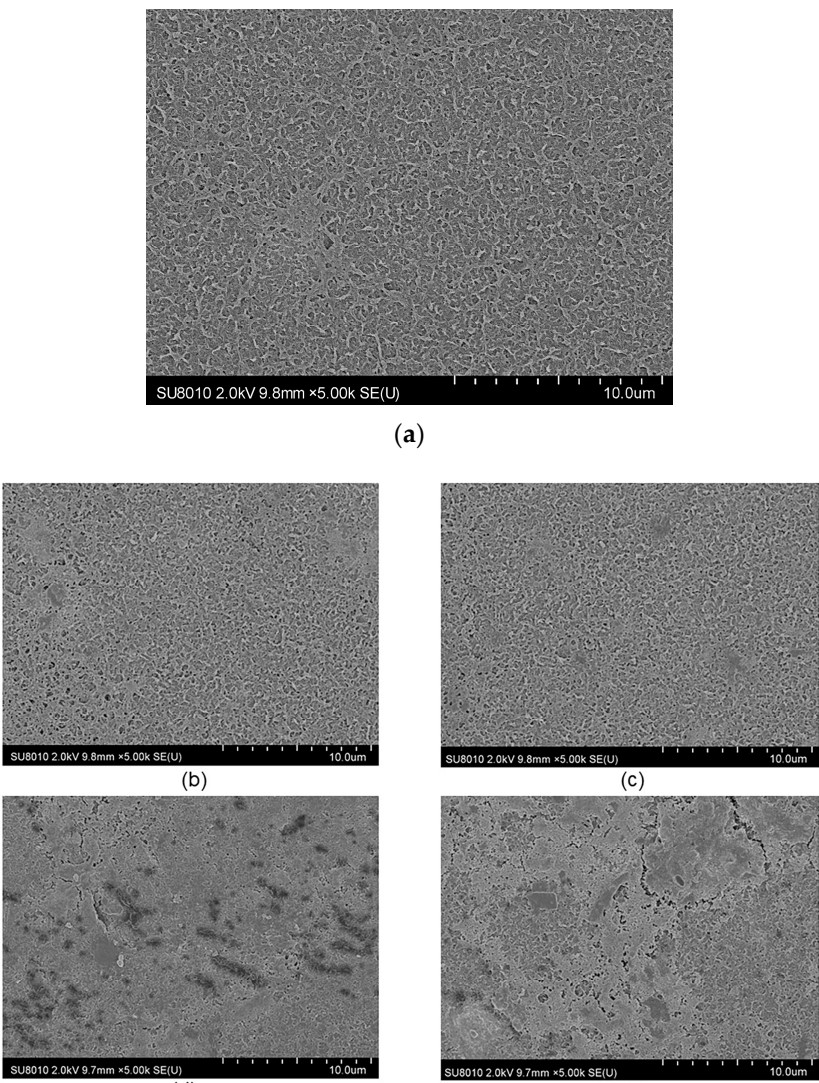

**Figure 8.** The effect of $Ca^{2+}$ or HA on RO membranes was characterized by SEM in the presence or absence of an electric field. (**a**) The unused RO membrane, (**b**) effect of $Ca^{2+}$ with no electric field, (**c**) effect of $Ca^{2+}$ with electric field, (**d**) effect of HA with no electric field, (**e**) effect of HA with electric field.

Figure 8b,c demonstrate the impact of $Ca^{2+}$ on the fouling of the RO membrane in the absence and presence of an electric field, respectively. It can be seen from the figures that the surface fouling of the membrane is relatively loose in the presence of an electric field, whereas the fouling is dense in the absence of an electric field. Given that the test water was artificially prepared, with the NaCl solution combined with only $CaCl_2$, it can be assumed that the pollutants are primarily $CaCl_2$. The SEM morphology analysis results align with the aforementioned observations and the increased electric field alters the trajectory of $Ca^{2+}$, making it less likely to be deposited on the surface of the film.

Figure 8d,e showcase the impact of HA on the fouling of the RO membrane in the absence and presence of an electric field, respectively. The pores of the membrane are

blocked by pollutants. However, increasing the electric field of the membrane leads to less dense pollutant deposition, making it easier to remove from the surface. Conversely, without the electric field, the fouling is denser and more resistant to cleaning. These findings are consistent with the filtration results, indicating that the electric field better inhibits and slows down membrane fouling.

## 4. Conclusions

In this study, the performance of a modified RO plant was evaluated by introducing an electric field into the system. The effects of different voltage levels on the RO desalination performance and membrane fouling were investigated. The results of the experiment illustrated that the use of an electric field beneficially impacted permeate flux, reduced the energy expenditure per unit of water produced, and abated membrane fouling compared to having no electric field, thus lengthening the membrane life.

The results of the experiments demonstrated that the impact of the electric field on solutions with concentrations of 1000, 2000, and 3000 mg/L became more pronounced with an increasing voltage magnitude. The highest voltage (25 V) increased the permeate flux by 1.02%, 1.23%, and 1.37%, respectively. Furthermore, by applying an electric field of 25 V to solutions with concentrations of 2000 and 3000 mg/L, the *SEC* was reduced by 31% and 59%, respectively. However, the solute rejection was not significantly improved by the use of the electric field.

Regarding membrane fouling, the electric field inhibited the decrease in permeate flux due to $Ca^{2+}$ contamination. Nevertheless, the final permeate flux reduction was similar to that without the use of the electric field. On the other hand, the effect of HA fouling on permeate flux was more pronounced, with the permeate flux decreasing from 5.49 to 4.81 $L/(m^2 \cdot h)$ after 120 h of operation. However, without an electric field, the permeate flux decreased to 4.23 $L/(m^2 \cdot h)$. Further studies are needed to determine the exact mechanism behind this phenomenon.

In conclusion, the results demonstrate that the use of an electric field can enhance the efficiency of RO treatment for brackish water and mitigate membrane fouling. This method of reducing the fouling of membranes and improving the efficiency of RO by incorporating electric fields has great potential for desalination and purification applications.

Moving forward, our research in this field will be directed toward treating high-salinity water while also considering the economic feasibility, thus establishing a foundation for the utilization of electric fields in RO membrane engineering.

**Author Contributions:** Conceptualization, C.F. and X.Y.; methodology, C.F. and X.Y.; formal analysis, C.F. and X.Y.; investigation, C.F.; resources, X.Y.; data curation, C.F.; writing—original draft preparation, C.F.; writing—review and editing, C.F. and Y.G.; visualization, C.F.; supervision, X.Y.; project administration, C.F.; funding acquisition, X.Y. All authors have read and agreed to the published version of the manuscript.

**Funding:** This research was partially funded by a grant from Fujun Li.

**Institutional Review Board Statement:** Not applicable.

**Informed Consent Statement:** Not applicable.

**Data Availability Statement:** The research data will be available upon request.

**Acknowledgments:** The authors are grateful to Fujun Li for his financial support.

**Conflicts of Interest:** The authors declare no conflicts of interest. The funders had no role in the design of the study; in the collection, analyses, or interpretation of data; in the writing of the manuscript; or in the decision to publish the results.

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
