# Peer review of "Effect of Electric Field on Membrane Fouling and Membrane Performance in Reverse Osmosis Treatment of Brackish Water"

_applsci, doi:10.3390/app14020575_

Round 1

Reviewer 1 Report

Comments and Suggestions for Authors

The manuscript titled “Effect of Electric Field on Membrane Fouling and Membrane Performance in Reverse osmosis Treatment of Brackish Water” and written by Caixia Fu et al. has some shortcomings that have to be resolved before being considered for publication. I recommend a major revision based on the following comments:

1.      Page 1, line 14. What are these flow rates in percentages? increases? Please clarify.

2.      Page 1, line 18, please, provide space between 120 and h. Revise the entire manuscript. (page 3, line 119, page 4, line 140…)

3.      Page 1, lines 27-28. The authors mentioned that RO has a small footprint, low operating cost, convenient management, and low energy consumption. Include that this is in comparison with other desalination technologies such as MSF or MED.

4.      The authors studied the inhibition of calcium fluoride fouling on a reverse osmosis membrane. This is a very interesting topic, however, in the introduction, the authors have not sufficiently reflected the impact of fouling by poorly water-soluble salts such as calcium sulphate or calcium carbonate in reverse osmosis desalination plants. There are several relevant studies that have evaluated the impact of fouling on the performance of these types of plants even when anti-scaling agents are used in the pre-treatment. The authors should include a paragraph citing some papers and commenting on the impact of fouling, which would enhance this manuscript. Some suggestions: Desalination 491, 114582; Membranes, 12(12), 1287; Membranes, 11(8), 616; Applied Sciences 10 (14), 4748; Membranes, 12(9), 894; Journal of Membrane Science, 579, pp. 52–69; Desalination and Water Treatment 73, 46-53.

5.      Page 4 line 140. The text in a different format.

6.      Page 5, line 167. The authors used the term desalination rate. This term is rare and not used in the literature. Please use solute rejection to avoid confusion for readers.

7.      In Equation 1, from where is the term 60, minutes? Please, clarify.

8.      Please, revise the equation 2, the terms used are rare and hard to understand, it is supposed that the solute rejection depends on the permeate concentration and feed concentration. Please, explain the terms “at the moment of T”, what is this?

9.      Eqs (5)-(7). Please, use the term specific energy consumption to have the things clearer for the readers.

10.  Page 8, Fig 4b. Why the increase of the voltage produced an increase of permeate flux? Was this observed in other studies? Did the authors measure the temperature? And the term membrane flux is not correct, it should be permeate flux. (water flux across the membrane).  

11.  Does section 3.5 make sense? I mean, the authors have fouled the membrane with calcium fluoride, there is no need to characterize the fouling as it is already known what it is.

Reviewer 2 Report

Comments and Suggestions for Authors

Article: Effect of Electric Field on Membrane Fouling and Membrane Performance in Reverse osmosis Treatment of Brackish Water

General comment: Authors put their best effort to write the article on effect of electric field on membrane fouling and membrane performance in reverse osmosis treatment of brackish water. The article can be accepted after minor revision. However, authors can include a table to compare their data with similar works done by other researchers in a section before conclusion.

Reviewer 3 Report

Comments and Suggestions for Authors

Overall, it is a good manuscript. Few changes are required to enhance the quality of the research conducted.

1. Some of the references seem to be outdated e.g., few from 2012. It is suggested to include the following for better readership. 

https://doi.org/10.1039/C8RA03810D

2. 2.3 Steps in the implementation of the experiment: please check the font and language mistakes. Similarly, material and methods section has few language mistakes. 

3. Add the cost calculations for the optimal voltage chosen. As addition of energy into the system will increase the treatment cost and may impact the scalability of this novel work. 
4. 3.6 Theoretical analysis of the action of electric fields on RO: this section is not reflecting any specific results rather literature is cited for general discussion. It is suggested to either provide mass transfer results or move this section with other relevant portions of the manuscript. 

Reviewer 4 Report

Comments and Suggestions for Authors

Desalination rate increase by< 1 % when electric field is used. Can authors justify the effect of use of electric field? Specially in case of Ca2+ 

Round 2

Reviewer 1 Report

Comments and Suggestions for Authors

The authors have adressed all my comments